# Preliminary Findings on How Different Management Systems and Social Interactions Influence Fecal Glucocorticoid Metabolites in White Rhinoceros (*Ceratotherium simum*)

**DOI:** 10.3390/ani12070897

**Published:** 2022-03-31

**Authors:** Leticia Martínez, Gema Silván, Sara Cáceres, Jose Manuel Caperos, Jesús Fernández-Morán, Miguel Casares, Belén Crespo, Paloma Jimena de Andrés, Juan Carlos Illera

**Affiliations:** 1Department of Animal Physiology, Veterinary Faculty, Complutense University of Madrid, 28040 Madrid, Spain; martinez.lett@gmail.com (L.M.); gsilvang@vet.ucm.es (G.S.); sacacere@ucm.es (S.C.); belencre@ucm.es (B.C.); jcillera@ucm.es (J.C.I.); 2Clinical Psychology Unit (UNINPSI), Department of Psychology, Comillas Pontifical University, Calle Mateo Inurria 37, 28036 Madrid, Spain; jcaperos@comillas.edu; 3Zoogical Area of Parques Reunidos Group, Casa de Campo s/n, 28011 Madrid, Spain; jfmoran2022@gmail.com; 4Bioparc Valencia, Avenida Pio Baroja 3, 46015 Valencia, Spain; miguel.casares@bioparcvalencia.es; 5Department of Animal Medicine and Surgery, Veterinary Faculty, Complutense University of Madrid, 28040 Madrid, Spain

**Keywords:** glucocorticoids, reproduction, management, stress, white rhinoceros

## Abstract

**Simple Summary:**

In recent years, interest in improving the welfare of wild species in captivity has grown. The aim of this investigation was to evaluate the effects of different social environments and management systems on the cortisol secretion of fourteen white rhinoceroses (*Ceratotherium simum*) living under different conditions by using use noninvasive methods. The fecal glucocorticoid metabolite secretion was found to be affected by both management systems and social interactions. Additionally, sex is another factor that seems to influence. This research provides a deeper understanding of glucocorticoid production in white rhinoceroses, but more studies are needed to fully understand its influence on reproductive biology.

**Abstract:**

White rhinoceroses (*Ceratotherium simum*) are the most social and gregarious species of all rhinoceroses known worldwide. One of the most critical effects of elevated glucocorticoid concentrations, especially in threatened species, is its relation to chronic stress, which could potentially lead to immunosuppression and reduced reproductive activity. Our aim is to determine how different social environments and management systems might be influencing the secretion of fecal glucocorticoids in white rhinoceroses. We have analyzed the concentration of fecal glucocorticoid metabolites in 658 fecal samples from 14 white rhinoceroses, seven free-ranging rhinos, and seven rhinos from two different managed captive populations. HPLC techniques were used to determine the main glucocorticoid metabolite found in this species, and a competitive EIA was used to establish fecal 5α-pregnan-3 3β, 11β, 21-triol-20-oneglucocorticoid metabolite (fGCM) levels. Our results reveal that management systems and social interactions had an influence on fGCM levels, suggesting that the more restrictive the management and social conditions are, the higher the glucocorticoid concentrations that are found. Additionally, sex was also found to influence fGCM levels, as in females, fGCM concentrations were higher than in males. We conclude that the analysis of glucocorticoids in relation to other factors is a powerful tool to assess adrenocortical response in white rhinoceros in order to broaden the knowledge of their reproductive biology and improve the management of the species.

## 1. Introduction

White rhinoceros (*Ceratotherium simum*) are classified as a Near Threatened species due to the increasing poaching rate affecting their populations [1]. Not only is the species at risk, but this also threatens the potential key role of this megaherbivore as a driver of savanna functioning [2]. The majority of the five different species of rhinoceroses that currently exist worldwide are classified as Critically Endangered (IUCN Red List). Translocation and re-introduction to protected areas may be the only means to safeguard the species [3]; thus, further research to help clarify the causes of their low reproductive rates in captivity is essential to ensure their survival [4]. 

White rhinoceroses are known to be the most social and gregarious of all the rhinoceros species [5], and management conditions play a very important role in their reproduction success and behavior [4]. In captivity, the complexity and variety of factors that affect their natural behavior frequently leads to reproductive failure, in addition to affecting animal welfare and social behavior [6].

According to Leader-Williams et al. [7], there are three categories of management systems related to rhino reproduction:-Captive breeding (captivity)—Rhinos in small or very small areas (<10 km^2^), with partial or total food supplementation, a high frequency of veterinary care intervention, and a manipulated reproduction system;-Semi-wild (semi-captivity)—Rhinoceroses in areas that are generally small (<10 km^2^), with partial food supplementation and high management intensity, but with a natural reproduction system. The animals live in an enclosure under human care, but the conditions seek to reproduce, as closely as possible, the conditions prevailing in their natural environment;-Wild (free-ranging)—Free rhinos in large areas (>10 km^2^) with no food supplementation, little veterinary intervention, and natural reproductive systems.

Captive animals tend to develop abnormal behaviors that might lead to stress, fear, or aggression, deteriorating their physical and psychological health [8,9,10,11]. Different hormones play a crucial role in controlling these behavioral and physiological processes; therefore, hormonal profiles have been used as effective monitoring tools in a variety of studies for conservation biology, wildlife management, and behavioral ecology, among other things [12,13,14,15].

Under a stressful situation, organisms undergo some changes to adapt to and deal with the new challenge. The release of cortisol is part of this physiological adaptation, an essential unspecific response for the organism to cope with stressful stimuli and survive [9,16,17,18]. Glucocorticoids, similar to cortisol, are crucial components in evaluating both acute and chronic stress responses, offering substantial information about how animals perceive and adapt to their surroundings [18,19,20].

A short-term physiological stress response can be detected by a significant increase in fecal glucocorticoid metabolites in diverse situations, such as the transportation/translocation of rhinoceroses [3,21,22,23], dehorning procedures [24], or the event of giving birth in other ungulates [25]. However, recent studies have concluded that dehorning is not related to long-term physiological stress in white rhinoceros [24]. Whilst an acute increase in circulating glucocorticoid levels enables an individual to deal with short-term stressors, chronic stress can have other negative impacts, as it is related to high glucocorticoid concentrations for long periods of time, and consequently may lead to immunosuppression and reduced reproductive activity [16,26,27].

Non-invasive glucocorticoid determination has provided substantial information for assessing stress responses and understanding how animals cope with and adapt to different stimuli or situations. In non-invasive studies, fecal samples are widely utilized and convenient, especially given their simplicity of use with wild populations [28]. Fecal glucocorticoids prevail as an essential component in evaluating the stress response [18] but have also been related to other aspects, such as social rank [20,29,30,31] and temperament [15]. Other parameters, such as age, environment, season, health issues, and specific individual behavioral patterns, have a strong influence, and should also be considered when possible [18,25,29,32,33,34].

Many studies in various species have also related glucocorticoid concentration to seasonality, finding in some a significant increase in glucocorticoid levels during winter [35,36] and in others, during summer [25,29]. However, different studies have not found any relation between the number of fecal glucocorticoid metabolites and the season [21]. For wild rhinoceroses, this seasonal difference is related to rainfall abundance (as a link to available forage and water resources), which is apparently connected to an increase in birth rates [37]. All authors reviewed agree on the importance of considering the many variables that could influence the hypothalamic–pituitary–adrenal axis, and hence, the secretion of glucocorticoids.

Assuming that the low reproductive rates found in captive white rhinoceroses could be related to long-term high glucocorticoid concentrations, our main hypothesis is that management systems and social interactions influence low reproductive rates by increasing glucocorticoid secretion. Therefore, the aim of this study is to determine how different management systems and social interactions may influence glucocorticoid production in white rhinoceroses. In an attempt to consider as many variables as possible related to their levels of fecal glucocorticoid metabolites, other factors such as sex, season, and social rank have been taken into account.

## 2. Materials and Methods

### 2.1. Animals 

A total of 658 fecal samples from 14 white rhinoceroses were collected from three different groups. Based on Leader-Williams et al.’s [7] management systems and rhino social interactions, the rhinos were divided into three groups:○Zoo Madrid (n = 2). This group consists of two white rhinoceroses—one female (“Female 1”, 15 years old) and one male (“Male 1”, 40 years old approximately) housed together in a daily outdoor area. These animals live in captivity under the following characteristics—enclosure of size <10 km^2^, total food supplementation, high frequency of veterinary care intervention, and manipulated reproduction system. Regarding social interactions, as these animals live in the same enclosure together with their young, they have intraspecific interactions. However, they are not able to establish any close inter-specific relations.○Bioparc Valencia (n = 4). This group includes two females (named “Female 2” and “Female 3”, 7 and 6 years old, respectively) and a male (“Male 2”, 20 years old) housed together in a daily outdoor area. An additional male (“Male 3”, 35 years old) is housed in a separate enclosure with olfactive, auditory, and visual contact with the other rhinos to try to stimulate natural sexual behavior in the group. This captive population lives in larger enclosures, but these are still smaller than 10 km^2^. The animals have total food supplementation, a high frequency of veterinary intervention, and a manipulated reproduction system after previous attempts to achieve gestation through natural copulation without success. Here the animals have wide social interactions as they establish intra- and inter-specific contact with typical animal species of the savannah (zebras and antelopes, among others).○South African Reserve (n = 8). This group consists of 8 white rhinoceroses at reproductive age or close to it, and variable individual conditions—5 females (“Female 4, 5, 6, 7, 8”, aged 30, 30, 10, 21, and 13 years old, respectively) and 3 males (“Male 4, 5, 6”, aged 7, 7, and 9 years old, respectively). All the females were accompanied by their young. This group of animals is kept under a management system that combines features of both semi-captivity and free-ranging management conditions, but since the aforementioned description is mainly based on breeding conditions, we will consider this last group as being in a free-ranging management system with the following characteristics—the animals live in a private reserve in South Africa of over 10,000 hectares (>100 km^2^) with many different animal and plant species, allowing the rhinoceroses to use their space and time freely. These animals rarely receive partial food supplementation (especially during winter) and occasional management, but they have a natural reproduction system and very little veterinary intervention (only in extraordinary circumstances). Animals receive eventual human care and little tourist/vehicle pressure, with a prevailing wild environment and a natural reproduction system. Animals roam in this large, enclosed area, where they establish relations of their choice among themselves (intra-specific) and with other typical African species (inter-specific). 

Keeper questionnaires were used to assess behavioral characteristics within the three groups. 

### 2.2. Sample Collection, Conservation and Lyophilization 

Since, in captivity, animals sleep in separate enclosures, fresh fecal samples from each individual were collected in the morning after defecation intermittently over a period of two years, especially in summer and winter at each location. Samples were labeled and frozen until further lyophilization. 

In the free-ranging population, samples were collected by patrolling the reserve before sunrise until finding the rhinoceroses of the study and waiting patiently for fresh samples. Each sample was collected, correctly identified, and labeled for each animal, and was then kept in a portable cooler box until being stored in a freezer for further lyophilization. Fecal samples in the wild were much harder to collect since we could not use any GPS tracker or localization device. Therefore, samples could not be collected on a strict daily basis for every free-ranging rhino included in the study. 

For lyophilization, each fecal sample was homogenized and introduced into the corresponding labeled vial. To freeze-dry the samples, we programmed a Telstar Lyoquest-85 for 48 h at 45 °C under a vacuum pressure of 0.010 mBar. After lyophilization, the samples were stored in vacuum conditions for future analysis.

### 2.3. Extraction

The extraction protocol for glucocorticoids was also performed at the University of Wien and followed their standardized method and guidelines [38,39].

Briefly, 0.2 g of dry feces was mixed with 0.8 mL of distilled water and 5 mL of methanol. Samples were vortexed for 30 min and centrifugated for 10 min at 1500× *g*. After centrifugation, 1 mL of the methanol was transferred into a new vial and evaporated under a gaseous nitrogen stream. Samples were then reconstituted in 1 mL methanol, and aliquots were kept frozen for subsequent analyses.

### 2.4. HPLC Analysis

Fecal glucocorticoid metabolites were analyzed using high-performance liquid chromatography (HPLC) to determine and confirm the main glucocorticoid metabolite occurring in this species. Several pools of fecal extracts were prepared. 

The HPLC protocol for fecal glucocorticoid metabolites was performed at the University of Vienna (Vetmeduni Vienna) following the chromatographic conditions described by Palme and Möstl [40]. In brief, several pools corresponding to the samples of both males and females of each group were made. In addition, two pools from a pregnant female (Marina) were included, one of the prepartum samples and another one of the postpartum samples. The samples were eluted with 5 mL of 80% methanol to remove potential contaminants. The HPLC used a C-18 column [Novapak C18 column (3.9 × 150 mm), Millipore Corporation, Milford, CT, USA] with a guard column (Mini-Guard column C18) connected to a fraction collector. Fecal samples were separated by a linear gradient from 50 to 75% methanol during the first 40 min transitioning to 100% methanol for up to 55 min. A total of 95 fractions were collected. The fractions obtained were reconstituted in phosphate buffer, and cortisol immunoreactivity was quantified using the EIA technique detailed below.

### 2.5. Hormonal Analysis

A competitive EIA developed by Schwarzenberger et al. [39] was used to analyze the concentrations of fecal glucocorticoid metabolite detected by HPLC. Briefly, 96-well polystyrene plates were coated with 250 µL of a 50-µg solution of protein A dissolved in 250 mL of coating buffer (carbonate/bicarbonate 0.05 M, pH 9.6). The plates were sealed and incubated at room temperature overnight. The plates were washed, and 300 µL of a second coating buffer (Tris PBS-BSA pH 7.5) was added. The plates were resealed and incubated overnight at room temperature.

The plates were washed, and 50 µL of EIA buffer (Tris-saline:BSA pH 7.5) was added to wells A1 and B1, which acted as reaction blanks, and to wells C1 and D1, which acted as maximum binding wells (B0). In consecutive wells, the 7 standards were added in duplicate (ranging from 500 pg/100 µL to 2 pg/100 µL). The remaining wells of the plate were coated with 50 µL of the test samples in duplicate. Subsequently, 100 µL of the antibody dilution was added to all wells except the blank wells, followed by 100 µL of the biotinylated steroid. The plates were incubated overnight at 4 °C with agitation. The plates were washed, and 250 µL of the conjugate solution (streptavidin-HRP) was added to all wells. The plates were incubated for 45 min at 4 °C. The plates were washed, and 250 µL of the substrate (TMB) was added, and this was then incubated for 45 min at 4 °C. The reaction was stopped with 50 µL of 20% sulfuric acid. The absorbance was measured on an EIA reader, and the hormone concentrations were calculated using specific software and expressed in ng/g feces. Assay sensitivity was 0.8 pg/well. Cross-reactivity with other steroids was: 5α-pregnan-3β, 11β, 21-triol-20-one 100%; 5α-pregnan-3β, 11β, 20β, 21-tetrol 110%; 5α-pregnan-3β, 11β, 17α, 21-tetrol-20-one 45%; 5α-androstan-3β, 11β-diol-17-one 230%; cortisol and corticosterone < 1%. The variation coefficients were 14% (intra-assay) and 9.1% (inter-assay). The linearity of the glucocorticoids in the serial dilutions of female and male rhino pools against the standard curve shows optimal linearity and parallelism between dilutions 1:10 and 1:100. 

As our intention was to assess the samples using a non-invasive procedure (which is especially important in threatened species), we opted for a physiological validation of fGCM. Thus, we analyzed the concentrations of fGCM during gestation in a pregnant female in order to obtain a concentration value that reflects a natural acute stress situation (giving birth) in a white rhinoceros.

### 2.6. Statistical Study

SPSS 25.0 (IBM Statistical Package for the Social Sciences, Armonk, NY, USA, 2017) was used for statistical analysis. Fecal glucocorticoid metabolite concentrations were processed as a continuous variable. For the descriptive analysis of this continuous variable, we used the mean value (ng/g of dry feces) and the standard error of the mean (S.E.M.). Group, sex, and season were processed as categorical variables to determine possible hormonal variations among the different animals. To assess the relationships of the variables sex, season, and group with hormonal levels, the mean values of each subject were analyzed using nonparametric statistics. The Wilcoxon test for paired samples was used to assess the effect of season. To assess the effect of sex, the U Mann–Whitney test for independent samples was used. To assess the effect of the group, a Kruskal–Wallis test was performed, correcting the significance level using the Bonferroni test when necessary. In an attempt to give greater weight to the study, since the number of animals included is low, a statistical significance level of *p* < 0.10 was established. Therefore, in all statistical comparisons, *p* < 0.10 was accepted as denoting a significant difference.

## 3. Results

### 3.1. Fecal Glucocorticoid Metabolite Identification by HPLC

The peak of the first glucocorticoid metabolite was observed at approximately 8 min; this peak occurred in all samples, and subsequently, another smaller peak was observed at 30 min (Figure 1). The fecal glucocorticoid metabolite that was identified was 5α-pregnan-3β, 11β, 21-triol-20-one (fGCM).

### 3.2. Fecal Glucocorticoid Metabolite Identification and Acute Stress

In order to establish a baseline for acute stress as related to fGCM levels for physiological validation, we analyzed the samples of one of the females (Marina, see Figure 1) that gave birth during the study. We found an fGCM peak (4502.7 ng/g) around the time of giving birth (Figure 2), which is a natural, stressful situation characterized by an elevation in circulating glucocorticoids. The fGCM concentrations of this female (Female 1) during the prepartum, partum, and postpartum stages were not considered in the general analyses. 

### 3.3. Fecal Glucocorticoid Metabolite Analysis: Analysis per Season

The main immunoreactive fecal cortisol metabolite identified in feces by HPLC was 5α-pregnan-3β,11β, 21-triol-20-one (fGCM). Then, the fGCM levels of all collected samples (n = 658) were analyzed by EIA. The variation in the fGCM levels of each individual between samples in summer and in winter was not constant, as in six animals, the fGCM levels were higher in winter than in summer, while in the other eight animals, the fGCM levels were higher in summer. Therefore, our results show no statistically significant differences in the fGCM mean concentrations between winter samples (622.68 ± 17.45 ng/g) and summer samples (715.85 ± 23.92 ng/g) (*p =* 0.826). In addition, the mean fGCM values for winter and summer were similar in all groups studied (Figure 3).

After this observation was made, the mean value of each animal was used for the remaining analyses, excluding the mean values at prepartum, partum, and postpartum in Female 1 (Table 1). 

### 3.4. Fecal Glucocorticoid Metabolite Analysis: General Analysis by Sex

In a general analysis, we found a statistical difference between both sexes. A significantly higher mean fGCM concentration was found in females (805.81 ± 20.01 ng/g), which was almost double the mean fGCM in males (522.87 ± 16.74 ng/g) (*p* < 0.081) (Figure 4).

### 3.5. Fecal Glucocorticoid Metabolite Analysis: General Analysis per Group

Our results show statistically significant differences in fGCM mean concentrations among the three groups (*p* = 0.094), with the fGCM mean for animals from Zoo Madrid (980.67 ± 37.16 ng/g) being statistically higher than that of the animals living in the South African Reserve (446.38 ± 27.86 ng/g) (*p* = 0.031) (Figure 5).

#### Variables Sex and Group

A similar variation in fGCM concentrations was observed in the three groups when the results were divided according to sex. The male (812.73 ± 61.65 ng/g) and female (1115.76 ± 34.43 ng/g) from Zoo Madrid had the highest means of fGCM, while the males (224.08 ± 22.15 ng/g; n = 3) and females (636.50 ± 126.24 ng/g; n = 5) in the South African Reserve had the lowest fGCM values (Figure 6) out of all the groups. Additionally, Figure 6 also shows that the values for the females are higher than those for the males living under the same management system and social conditions.

### 3.6. Social and Behavioral Observations

In this section, a descriptive analysis of the most important social and behavioral observations within different groups of males or females has been carried out in order to evaluate a possible relationship between the individual mean fGCM concentrations and the behavioral observations that were recorded.

In Bioparc Valencia, samples from two males and two females were assayed. Within males, Male 2, who was housed with two females in an outdoor daily area, showed the highest fGCM concentration and the highest variability out of the Bioparc males. To stimulate sexual behavior and female mating choice, Male 3 was housed in a different enclosure with olfactive, auditive, and visual bidirectional contact with the group. Although mating displays in the group were recorded, none of them led to a successful gestation during our study. A certain dominance of one of the females, Female 2, over the other (Female 3), including over the male that lives with them (Male 2), was observed. However, both females had similar fGCM levels.

For the South African Reserve, our results showed similar fGCM concentrations within males, but a wide range of mean fGCM values among females. It is remarkable that the mean fGCM of Male 6 (the dominant male) (200.63 ± 12.57 ng/g) was lower than that of Male 4 (268.38 ± 19.16 ng/g). Male 5, who started showing territorial behavior by the end of our study, had a similar mean fGCM (203.25 ± 13.03 ng/g) to the dominant male. Within females, Female 7 had the highest variability in fGCM levels (797.88 ± 156.05). This female had a more solitary behavior than the rest of the females studied.

## 4. Discussion

This study investigated the production and secretion of glucocorticoids in white rhinoceroses using non-invasive methods and revealed some remarkable insights into how social interactions, management systems, and sex influence fGCM secretion in these animals. This study is timely given the growing interest in finding non-invasive methods to assess possible causes of reproduction failure in the captive white rhino population. Furthermore, these non-invasive methods could also be used to assess welfare both in captivity and in the wild. 

In this study, the number of animals is relatively low, which limits the significance of the results. Since it is a remarkable fact that it is very difficult to find large numbers of a wild species with the behavioral and housing requirements of the white rhinoceroses sharing the same enclosure, many previous studies involving wild species have used a low number of subjects and produced meaningful and insightful findings [41,42,43,44]. A common practice is that, to increase the number of individuals in a study of wild animals, data from animals housed in different institutions are used [45,46]. However, given the design and aim of our study, this could not be performed here. Despite the limitation of the low number of animals included in the study, we think that our results have significant implications that could help to improve the management of ex situ white rhino populations, and, in the long term, the conservation of this species.

Our results show that fGCM concentrations might be related to the size of the enclosure and the composition of the herds since, in captivity, animals under the most restricted conditions (with smaller enclosures, smaller groups of animals, and no interspecific interactions) showed the highest fGCM concentrations. The effect of enclosure size has been studied in other captive species [47,48,49]; however, only one study has addressed this effect in rhinos. In this previous study, the area of the enclosure was not correlated with fecal glucocorticoid excretion in black rhinos [6]. However, we think that this factor might have influenced our results. On the other side, not only the size, but also the design of the enclosure, might be influential, since this same study found that stress levels were highest in rhinoceroses maintained in enclosures that allowed visitor viewing around a greater portion of their perimeters. However, this enclosure characteristic was not considered in our study. 

Given that the animals in captivity (Zoo Madrid and Bioparc Valencia) showed higher fGCM concentrations than animals in free-ranging conditions (South African Reserve), it seems that not only factors determining the management system but also other factors, such as the effects of visitors, might influence cortisol secretion in white rhinos. Other researchers have also detected increases in fGCM levels in captive animals compared to free-ranging populations; for example, in the Canadian lynx (*Lynx canadiensis*), and in wild dogs (*Lycaon pictus*) [50,51]. The elevation in cortisol as a response to acute stressors only lasts for a short period of time (which depends on the species), but the samples in our study were intermittently taken over a period of two years in all groups; therefore, the differences found in the different groups might be the result of other factors acting as long-term stressors in every group, such as the effect of the continued presence of visitors. Therefore, the lower fGCM concentrations in our free-ranging group would be expected, as the pressure of tourists had been previously documented to act as a stressor for captive animals in other species [52,53,54]. 

On the other hand, there is some controversy as to whether cortisol follows a seasonal pattern. It has been previously reported that cortisol secretion follows a seasonal pattern in captive Indian rhinos (*Rhinoceros unicornis*) [41]. However, our results reveal a minimally elevated mean fGCM value in summer samples compared to that of winter samples (not statistically significant); furthermore, the fluctuations in the values between summer and winter were different among the animals. Similarly, Brown and colleagues [45] found no evidence of seasonality in male black and white rhino sex steroid hormones, based on fecal androgen profiles. The authors also reported the independence of cortisol secretion from climatic variables in captive Asian elephants [42]. Therefore, we believe that climatic conditions may not have an influence on cortisol secretion in white rhinoceroses.

Additionally, since it has already been documented that maintaining high levels of glucocorticoids for long periods of time may lead to reproductive failure [6,16,26], so the elevated glucocorticoids found in the rhinos kept in captivity in our study might explain the low reproductive rates previously described in captive white rhinoceroses [4,55]. These results imply, therefore, that stressors associated with captivity are potential causes of chronic stress in white rhinos and may contribute to reproductive problems. This fact, in the long term, could be contributing to the captive population’s self-sustainability problem. It is desirable that future studies further investigate the number of sexual hormones in conjunction with glucocorticoids in white rhinos to assess the extent and influence of this finding on reproduction.

In the three groups studied, we found that females presented higher mean fGCM concentrations compared to males. Sex differences in adrenocortical activity have already been found in other mammals [46,56,57]. It has been hypothesized in other species that the difference may be due to an evolutionary adaptation of females to increase vigilance to protect and rear young and to avoid aggression from dominant males [46]. In our study, females from Zoo Madrid and from the South African Reserve lived accompanied by their calves. This might explain why their means are so much higher than those of males. However, in Bioparc Valencia, there were no calves, and even so, the females showed higher fGCM concentrations. This may reflect underlying differences in steroid metabolism and excretion routes [58,59], and in responsiveness to pituitary hormones between sexes [50,60,61]. Chronic stress is associated with the maintenance of high glucocorticoid concentrations for long periods of time and can lead to reduced reproductive activity [16], but the variety of mechanisms by which stress impacts reproduction may differ among sexes, since it can be influenced by the action of sexual steroids [26]. 

The composition of captive white rhino groups has a significant influence on social interactions between the animals [43]. In Bioparc Valencia, Male 2 (housed with the two females) showed the highest mean fGCM concentration and the highest variability out of the Bioparc males. His elevated level of fGCM could be associated with sexual reproductive behavior and direct daily contact with females. According to Carlstead and Brown [6], social behaviors related to mate choice may be associated with adrenocortical activity through inhibitory or stimulatory pathways, which probably have physiological impacts on reproduction in this species. To stimulate sexual behavior and female mating choice [44,62,63], Male 3 was housed in a different enclosure with olfactive, auditive, and visual bidirectional contact with the group. Although mating displays in the group were recorded, none of them led to a successful gestation during our study. According to Creel [64], whilst in the wild, it is more common to find elevated glucocorticoid concentrations in the dominant individual; in captivity, subordinate individuals tend to have higher glucocorticoid concentrations. Thus, our results strongly suggest that Male 3, being the dominant male, was not only due to the difference in fGCM concentration, but also due to his older age and recorded behavior, which was influencing and maybe even suppressing the reproductive behavior of Male 2 [31,65]. An elevation in fGCM levels has been related to dominance and social rank as an adaptation mechanism in some species that suppresses the reproductive activity of the subordinates or lower-ranked individuals. This strategy is species-specific and depends on other factors such as environmental conditions, breeding system, and how social rank is obtained and maintained [29,30,31]. 

In the South African Reserve, we found the highest variability in fGCM values. This could be due to the fact that free-ranging animals need to deal with daily challenges and short-term stressors (food and water availability, social interactions, health, shade, etc.), and the method of adapting to cope with every stressor is different in each individual [23,66]. Within the group of males, the youngest male (Male 4) showed higher levels of fGCM than the other individuals. Other studies, primarily in domestic and laboratory animals, have found that age and early experience can affect stress responses [67] since the experienced animal appears to adapt better to daily stressors. Male 5 began to show territorial behavior at the end of our study. His mean fGCM concentration was lower than that of Male 4, as if he were also a dominant male (with a minimal difference in his fGCM mean compared to Male 6, the major dominant). Similar to what we found in Bioparc Valencia, in this habitat, the dominant male showed lower fGCM concentrations than the subordinate males. In the species where dominance involves a high energetic maintenance cost and constant fighting, high-rank individuals are expected to have elevated glucocorticoid concentrations [20,64,68]. However, in the white rhinoceros, it seems that fGCM levels are not strictly linked to dominance, probably because this species is not aggressive [5]. It is also important to keep in mind that the concentration of fGCM in these males might be influenced by other factors, such as the presence of females in the group and its relation to reproductive behavior [6], and younger animals might deal with stress in a different way to older individuals [67].

In summary, the activation of the hypothalamic–pituitary–adrenal axis is an essential part of the adaptation response regulating glucocorticoid secretion in response to various physiological factors and environmental variables [16,18,21]. Therefore, when analyzing glucocorticoid metabolites and interpreting results, it is essential to cover as many internal and external variables as possible [15,18,25,31,33,34]. With the resources we had available for this study, we could only include the following variables related to glucocorticoid secretion: sex variation, individual differences, social environment, varying seasons, and the influence of different management systems. Our findings need to be further investigated by increasing the number of rhinoceroses and broadening the management conditions and housing facilities. Additionally, we should include any other variables involved in fGCM secretion in order to improve the general knowledge of rhinos’ reproductive biology and advance the conservation of this threatened species.

## 5. Conclusions

White rhinoceros are threatened with extinction from many causes, such as poaching or habitat loss. As poaching reaches unprecedented levels, a healthy and self-sustaining captive population of white rhinos that can serve as a safety net against extinction has become the most important target in ex situ conservation plans. However, the existing captive white rhinoceros population is far from self-sustaining because breeding success in captivity is poor. The secretion of cortico-adrenal hormones is controlled by a variety of internal and external factors that play an important role in the corticotrope axis and in the reproductive physiology of different species. This study was undertaken to investigate whether different specific management systems and social interactions may be indicators of chronic stress (as reflected by persistently elevated fecal glucocorticoid metabolite levels). In the relatively small number of animals in our study, higher fGCM levels appeared to be related to more restrictive management systems and social interactions. It is important to remember that the secretion of glucocorticoids can also be influenced by other variables, such as sex, tourist exposure, weather events, climate, animal–human interaction, etc. In our study, we found that fGCM concentrations were related to sex but not to the season, supporting a non-strict seasonal glucocorticoid secretion pattern in this species based on fecal glucocorticoid concentrations. We can conclude that the analysis of fecal glucocorticoids is a powerful tool to assess the adrenocortical response to different situations in white rhinoceroses. There is an important need to increase the number of cases in the statistical study in order to be able to extrapolate the data obtained to the general population. We believe that since the sampling is simple and the technique is affordable, the measurement of fecal glucocorticoid metabolites could be performed routinely in captivity in order to assess levels of stress in certain animals, especially if stress is negatively affecting the general welfare or the reproductive physiology of this threatened species.

## Figures and Tables

**Figure 1 animals-12-00897-f001:**
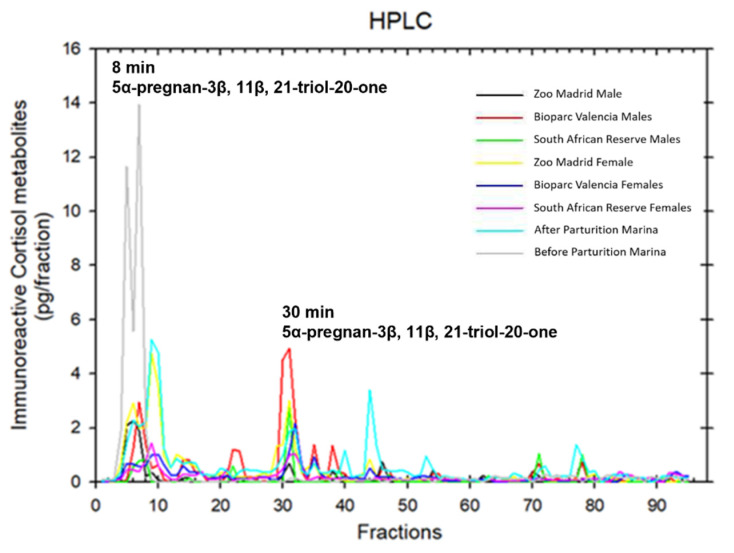
HPLC analysis of fecal glucocorticoid metabolites. Different colored lines refer to the different pools of samples prepared for HPLC that are specified.

**Figure 2 animals-12-00897-f002:**
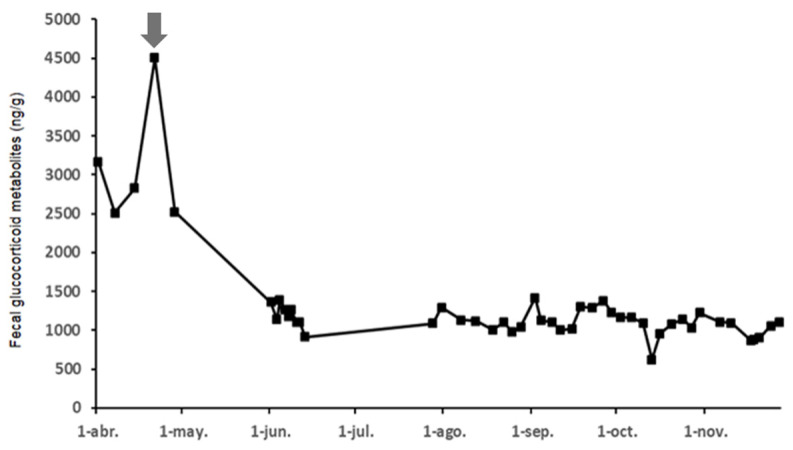
Hormonal profile of fGCM (ng/g) before, during and after giving birth in Female 1. The arrow points to the moment of partum. The hormonal peak was reached just before parturition (fGCM = 4502.7 ng/g).

**Figure 3 animals-12-00897-f003:**
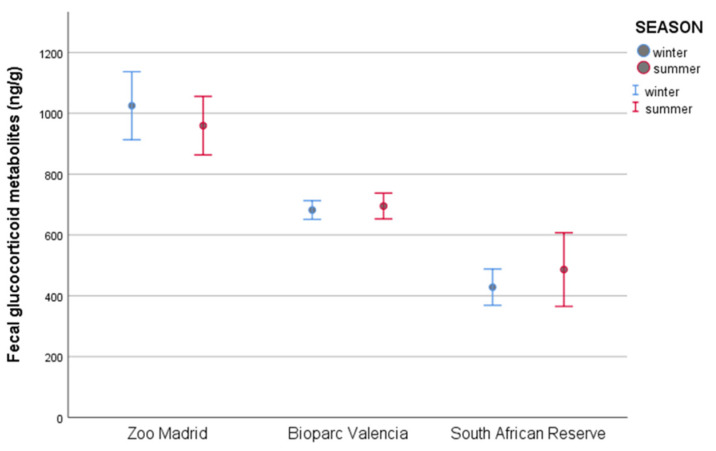
Fecal glucocorticoid metabolite concentrations (fGCM) ± standard error of the mean (S.E.M.) between seasons for animals from different groups.

**Figure 4 animals-12-00897-f004:**
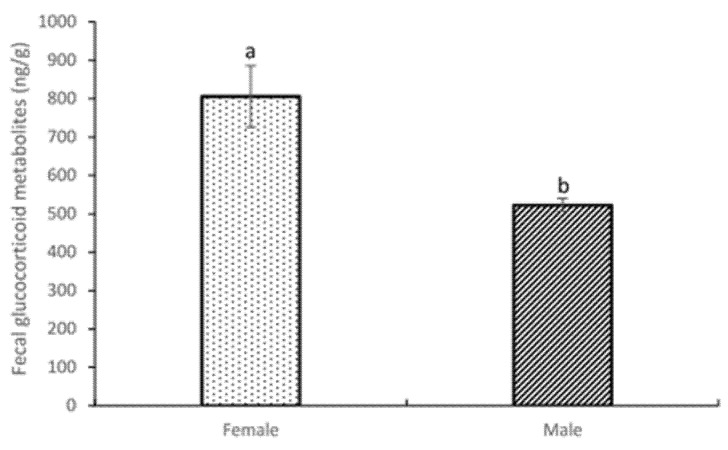
Fecal glucocorticoid metabolite concentrations (fGCM) ± standard error of the mean (S.E.M.) between different sexes. Values with distinct superscript letters (a, b) denote statistical differences (*p* value < 0.10) among groups using the U Mann–Whitney test.

**Figure 5 animals-12-00897-f005:**
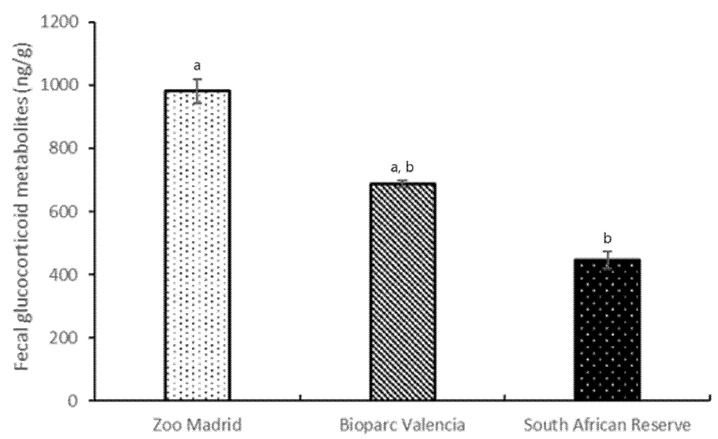
Fecal glucocorticoid metabolite concentrations (fGCM) ± standard error of the mean (S.E.M.) among the three groups studied. Values with distinct superscript letters (a, b) denoting statistical differences (*p* value < 0.10) among groups using the Kruskal–Wallis and Bonferroni tests.

**Figure 6 animals-12-00897-f006:**
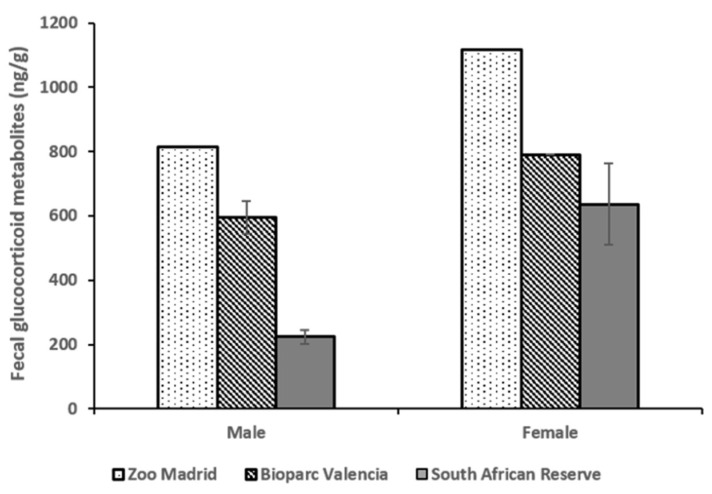
Fecal glucocorticoid metabolite concentrations (fGCM) ± standard error of the mean (S.E.M.) between the different sexes in the three groups.

**Table 1 animals-12-00897-t001:** Details of management characteristics, social interactions, sex, age (years) and fecal glucocorticoid metabolite concentrations (fGCM) ± standard error of the mean (S.E.M.) determined by EIA in the fourteen white rhinoceroses included in the study.

Study N°	Group		Management Characteristics			Social Interactions	Sex	Age	Mean ± S.E.M. fGCM (ng/g Dry Feces)
		System	Enclosure	Food Supplementation	Breeding	Inter-Specific Relations			
1	Zoo Madrid	Captive	Small	Total	Manipulated reproduction system	Absent	Female 1	15	1115.76 ± 34.43 *
2							Male 1	40	812.73 ± 61.65
3	Bioparc Valencia	Captive	Medium	Total	Manipulated reproduction system	Present	Female 2	7	790.17 ± 23.13
4							Female 3	6	790.11 ± 31.14
5							Male 2	20	645.44 ± 18.32
6							Male 3	35	545.45 ± 13.62
7	South African Reserve	Free-ranging	Large	Partial	Natural reproduction system	Present	Female 4	30	989.12 ± 62.75
8							Female 5	30	388.93 ± 37.65
9							Female 6	10	692.50 ± 46.98
10							Female 7	21	797.88 ± 156.05
11							Female 8	13	314.07 ± 40.32
12							Male 4	7	268.37 ± 19.16
13							Male 5	7	203.25 ± 13.03
14							Male 6	9	200.63 ± 12.57

* Excluding values during the prepartum, partum and postpartum stages.

## Data Availability

The data that support the findings of this study are available from the corresponding author upon reasonable request.

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
