# Peer review of "Preliminary Findings on How Different Management Systems and Social Interactions Influence Fecal Glucocorticoid Metabolites in White Rhinoceros (Ceratotherium simum)"

_animals, 2022, doi:10.3390/ani12070897_

Round 1

Reviewer 1 Report

The writing is clear.  However, the design was not appropriate even with the caveat of limited animals available for study.  The title and conclusions are not substantiated by the results. 

Reviewer 2 Report

HPLC Analysis;  I am thinking it will be good to added the detailed HPLC method in section 2.4.  Secondly, figure 1 (HPLC analysis) showing different colors can you add caption to denote what each of these pooled samples were? For example is the highest grey line pooled stressed rhino samples?  

Without the caption distinguishing the nature of each of the pooled it makes it difficult for the reader to know why there is so much between pooled sample variation in the immunoreactivity measured (y-axis). 

Thirdly, can you provide the retention time for 3H-Corticosterone on the graph (figure 1)?

Round 2

Reviewer 1 Report

The manuscript should be altered. The correlation between two individuals is of little vale; the animals values do not reflect changes due to physiological status but rather daily fluctuations. What hypothesis is tested with those results?. Likewise, the data for the individual around parturition and during the ensuing months demonstrate that you were able to measure responses that varied in a predictable manner with physiological state. Neither figure is adding value to the experiments purpose. The data for the experiment are presented in the table. You have made a liberal interpretation of those results after statistical analysis. 

Round 3

Reviewer 1 Report

Thank you for your careful consideration of recommendations.  The manuscript is well-written and provides information useful for the wildlife community.